# The type of diet consumed during prepuberty modulates plasma cholesterol, hepatic LXRα expression, and DNA methylation and hydroxymethylation during adulthood in male rats

Ana Aguilar-Lozano[1,2], Berenice Palacios-González[3]*, Martha Guevara-Cruz[1], Armando R. Tovar[1], Alam Palma-Guzman[4], Lilia G. Noriega [1]*

1 Department of Nutritional Physiology, National Institute of Medical and Nutritional Sciences "Salvador Zubirán", Mexico City, Mexico, 2 Biomedical Sciences PhD program, National Autonomous Mexican University (UNAM), Mexico City, Mexico, 3 Healthy Aging Unit from the National Institute for Genomic Medicine, Aging Research Center, Mexico City, Mexico, 4 Histology Laboratory from the XXI Century National Medical Center–Social Security Mexican Institute, Mexico City, Mexico

* bpalacios@inmegen.gob.mx (BPG); lilia.noriegal@incmnsz.mx (LGN)

**Data Availability Statement:** The data underlying the results presented in the study are available at

## Abstract

Childhood obesity increases the risk of developing metabolic diseases in adulthood, since environmental stimuli during critical windows of development can impact on adult metabolic health. Studies demonstrating the effect of prepubertal diet on adult metabolic disease risk are still limited. We hypothesized that a prepubertal control diet (CD) protects the adult metabolic phenotype from diet-induced obesity (DIO), while a high-fat diet (HFD) would predispose to adult metabolic alterations. Sprague-Dawley male rats were fed either a CD or a HFD during the prepubertal period (day 30–40 of age) and subsequently a chronic HFD or CD, respectively, until adulthood (day 220 of age). As controls, rats aged 30 days were exclusively fed a CD or a HFD until adulthood. Body weight and composition, metabolic rate, biochemical and hormonal plasma measurements, hepatic gene expression and methylation and hydroxymethylation levels were analyzed at ages 30, 40 and 220 days. The prepubertal CD prevented fat mass accumulation, lean mass loss and metabolic inflexibility, showed lower insulin, leptin and cholesterol concentrations in adulthood despite the chronic HFD. Notably, the prepubertal CD led to higher hepatic *Lxrα* expression, lower hepatic global DNA methylation and higher hydroxymethylation in adulthood despite a chronic HFD. Conversely, a prepubertal HFD decreased adult metabolic flexibility, increased serum cholesterol, and decreased *Lxrα* expression and global DNA hydroxymethylation, while also increasing DNA methylation levels despite a chronic CD. In summary, a prepubertal CD protected the adult metabolic phenotype from high cholesterol concentrations associated with increased hepatic *Lxrα* expression and lower hepatic global DNA methylation in adulthood, despite exposure to a chronic HFD. Conversely, a prepubertal HFD altered the adult metabolic phenotype.

the following link https://doi.org/10.6084/m9.figshare.25806877.

**Funding:** This research was funded by "Convocatoria 2023 para el fondo de apoyo a proyectos de investigación en el campo de la salud del Instituto Nacional de Ciencias Médicas y Nutrición Salvador Zubirán" to LGN. There was no additional external funding received for this study. The funders had no role in study design, data collection and analysis, decision to publish, or preparation of the manuscript.

**Competing interests:** The authors have declared that no competing interests exist.

## Introduction

Childhood obesity is known to increase the risk of developing obesity and metabolic disturbances like atherosclerosis in adulthood [1, 2]. The "Developmental Origins of Health and Disease" (DOHaD) hypothesis stipulates that the environmental stimuli during critical developmental windows can impact adult metabolic health [3]. Nevertheless, studies specifically investigating the effect of the prepubertal window of development on adult disease risk are still limited [4].

Studies have shown that a high-fat diet (HFD) and excessive weight gain during prepuberty in rats result in excessive fat mass accumulation and metabolic disturbances in adulthood, including increased levels of leptin [5, 6]. This programming of the adult metabolic phenotype during prepuberty may be driven by epigenetic mechanisms, such as DNA methylation and hydroxymethylation, which can have long-term effects on gene expression [3, 7–9]. Moreover, the transition from childhood to puberty, specifically the prepubertal period, is characterized by DNA methylation modifications in humans. These modifications can be influenced by environmental stimuli such as diet [10–15], and occur at gene promoters or genomic elements of peripheral blood cells, associated with lipid metabolism, as well as other metabolism-related genes such as insulin-like growth factor 2 (*Igf2)* [13] and propiomelanocortin (*Pomc*) [16]. Alterations in IGF2 function decrease muscle mass and glucose uptake, modifying metabolic flexibility [17], which is decreased during obesity and high-fat overfeeding, and therefore, the ability to switch fuels during feeding and fasting or upon changes in macronutrient composition is partially lost [18].

In murine models, DNA demethylation events in liver occur in enhancers harboring motifs for transcriptional factors involved in hepatic lipid metabolism [19]. Failure in demethylation during early life has been associated with increased cholesterol concentrations in adulthood [19]. Moreover, supplementation with folic acid, which enables methyl group donation to DNA, decreased hepatic proliferator activated receptor alpha (*Pparα*), acyl-CoA oxidase (*Acox1*) and carnitine-palmitoyl transferase (*Cpt1a*) gene expression [20]. Meanwhile, protein restriction during prepuberty, which corresponds to 30–40 days of age in male rats, resulted in hormonal alterations, including lower testosterone levels, in adulthood [21]. Both events could be led by changes in DNA methylation or hydroxymethylation levels. Furthermore, a maternal HFD induced hepatic global DNA methylation alterations [22]. Similarly, a maternal high-fructose diet resulted in increased liver X receptor-alpha (*Lxrα)* promoter methylation in the liver of the adult progeny. This led to increased levels of hepatic and serum cholesterol in adulthood, despite being fed a control diet (CD) until adulthood [23]. Interestingly, an adult HFD or high-fructose diet has also caused DNA methylation changes, like increased hepatic NADH:Ubiquinone oxidoreductase subunit B9 (*Ndufb*9) [24] or *Pparα* methylation [25], respectively.

However, it remains uncertain whether the type of diet during prepuberty, a critical plasticity window during development, can program the adult phenotype through epigenetic mechanisms, such as DNA methylation. Additionally, studies analyzing the prepubertal period as a critical window during development focus on how dietary interventions can condition the adult phenotype. However, there are no studies that evaluate whether a CD during this period could prevent the adult metabolic phenotype from the development of diet-induced obesity (DIO), and if so, what the potential mechanisms of this prevention could be. This information could enable new targets or strategies to ameliorate obesity prevalence in the adult population.

Therefore, this study aimed to understand whether a prepubertal CD or HFD could impact the adult metabolic phenotype. Additionally, we evaluated the impact of the different prepubertal diets on hepatic lipid accumulation and gene expression of enzymes and transcription

factors from the cholesterol metabolic pathway in adulthood. Finally, considering the stability of DNA methylation and hydroxymethylation status establishment, we determined the effect of the prepubertal diet on global hepatic methylation and hydroxymethylation. We hypothesized that a prepubertal HFD alters the adult metabolic phenotype, including fat mass accumulation, metabolic flexibility, lipid profile, and hepatic lipid accumulation, while a CD during prepuberty protects the adult metabolic phenotype from DIO.

## Methods and materials

### Rat feeding studies

Sprague-Dawley male rats (UPEAL-Cinvestav, Mexico City), age 30 days, were randomly assigned to four groups, housed in polycarbonate rat cages, three rats per cage, 12:12 light cycle, at ~22˚C and with free access to food and water. Groups were fed either a CD or a HFD for 180 days (**CC** or **HH**, respectively), or either a CD during prepuberty (Day 30–40 of age) and a subsequent chronic HFD until adulthood (**C→HFD**) or HFD during prepuberty (Day 30–40 of age) and a subsequent chronic CD until adulthood (**HFD→C**) (**Fig 1A**). Table 1 specifies the ingredients of the animal diets. We used AIN-93M formula as CD, where 9% of the energy corresponded to unsaturated fat. In the HFD, we added lard to reach a fat content that corresponded to 45% of the total energy. We obtained samples from six rats at postnatal day (PND) 30 (T0) for basal measurements. At PND 40 (T1), we collected samples from twelve rats, six rats fed with CD and six rats fed with HFD, to evaluate the effect of diet during the prepubertal period. Testosterone levels were assessed to ensure the prepubertal status of the animals (S1 Fig). Finally, at PND 220 (T2), we obtained samples from 6–5 rats per group to evaluate the impact of the prepubertal diet on adulthood. All rats were euthanized using isoflurane anesthesia, and the absence of reflexes was confirmed. The rats were fasted for 6 hours prior to the procedure. The sample size was calculated using the equation for independent samples. We collected samples, including plasma and liver, which were immediately flash-frozen in liquid nitrogen and stored at -80˚C until the corresponding experiments were performed. Additionally, hepatic tissue was collected in 10% formalin for histological analyses. Body weight and food intake were assessed once a week and every 2 days, respectively. Body composition was determined monthly by magnetic resonance (EchoMRI$^{TM}$). All experimental procedures performed were approved by the Animal Committee of the National Institute of Medical and Nutritional Sciences, Mexico City (Protocol FNU-1994-20/22-1).

### Energy expenditure measurements

We performed indirect calorimetry using metabolic chambers that assessed $O_2$ consumption and $CO_2$ production. The instrument used was a Comprehensive Lab Animal Monitoring System (Oxymax-CLAMS, Columbus Instruments), which was kept in constant room temperature. We placed each rat individually at separate chambers with constant air flow for 42 h, and calculated the volume of $O_2$ consumption ($VO_2$) and the volume of $CO_2$ production ($VCO_2$). We also calculated the respiratory exchange ratio (RER) as $VCO_2/VO_2$. The initial 6 h were designated as an adaptation period. Thus, the measurements obtained during that time were excluded from the analysis. The following measurements were taken with *ad libitum* feeding with the original diets (CD or HFD). Food was available from 14:00–7:00 h. Subsequently, food was removed from all chambers, and fasting measurements were taken from 08:00–17:00 h. Finally, to determine metabolic flexibility, we provided from 18:00–7:00 h the CD used throughout the study to all rats, to assure they were receiving a diet with the same amount of carbohydrates and lipids. Measurements were taken every 21 min. Metabolic flexibility was analyzed by calculating the ΔRQ, which displayed the difference between the average of the

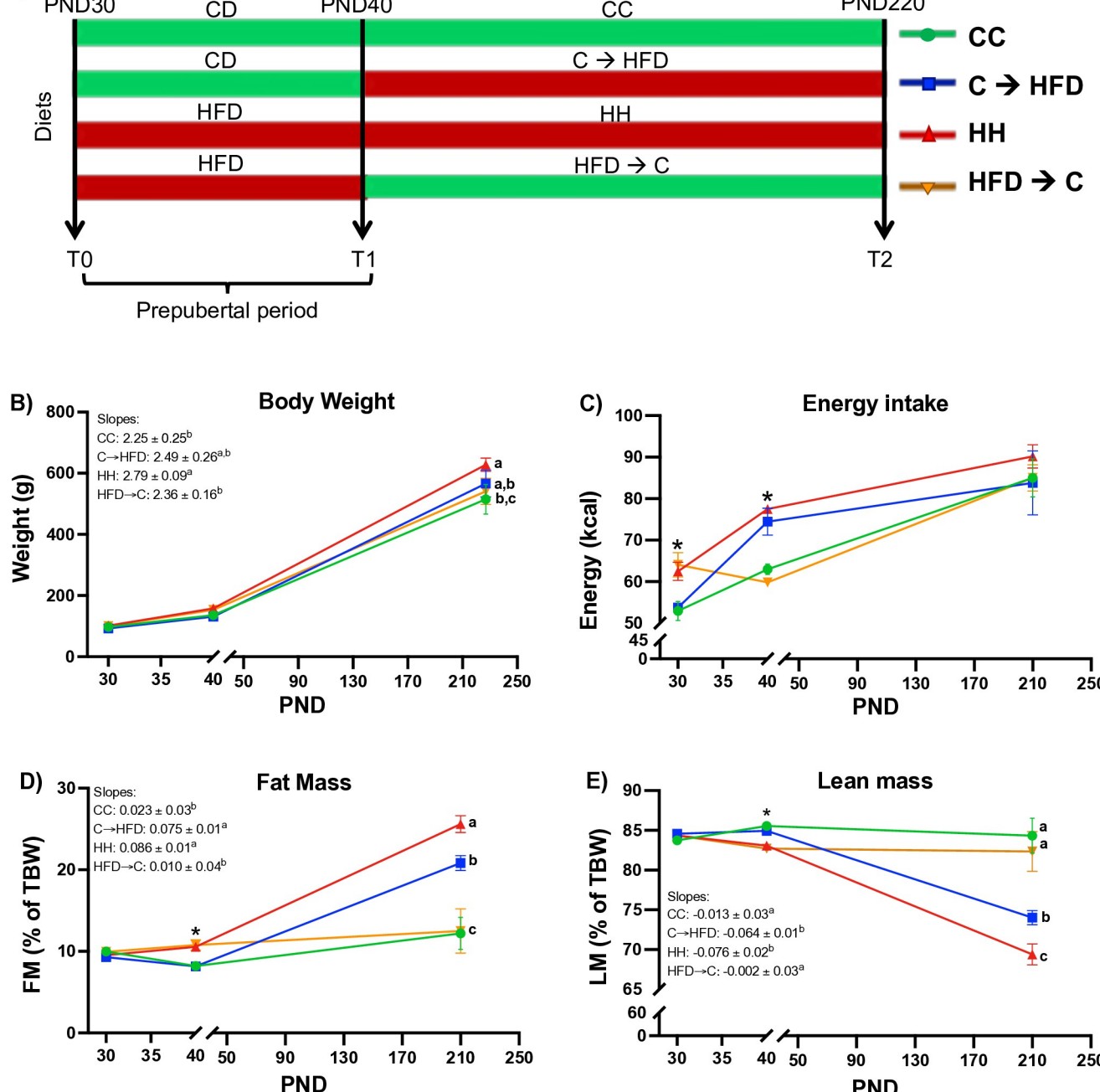

**Fig 1. A prepubertal CD decreases fat mass accumulation and prevents lean mass loss in adulthood despite a chronic HFD in Sprague-Dawley male rats.** *(A)* Study design, *(B)* body weight, *(C)* energy intake, *(D)* fat and *(E)* lean mass. Values are means ± SEMs (*n* = 5–6 rats per group in all panels). Mixed effects ANOVA analyses were performed followed by Tukey´s post hoc tests. Letters indicate significant differences among groups, where a > b > c ($p <$ 0.05) and *: $p < 0.05$ comparing CD vs. HFD in T1. Abbreviations: PND = Postnatal Day; CD = Control Diet; HFD = High Fat Diet; T0 = PND30, T1 = PND40, T2 = PND210; CC = Control Diet for 180 days; C→HFD = Control Diet for 10 days and HFD for 180 days; HH = HFD for 180 days; HFD→C = HFD for 10 days and CD for 180 days; FM = Fat mass; LM = Lean mass; TBW = Total body weight.

**Table 1. Composition of the different diets provided to Sprague-Dawley male rats.**

| Ingredients (g/kg) | Control Diet | High Fat Diet |
|---|:---:|:---:|
| Corn Starch | 466 | 220 |
| Casein | 140 | 175 |
| Granular Sugar | 100 | 100 |
| Dextrin | 155 | 100 |
| Vegetal Oil | 40 | 70 |
| Vitamin Mix | 10 | 12 |
| Mineral Mix | 35 | 42 |
| L-Cystine | 1.8 | 3 |
| Choline | 2.5 | 3 |
| Cellulose | 50 | 50 |
| Lard | - | 170 |
| Caloric Value (kcal/kg) | 3810 | 4810 |

RER taken during the four initial hours of fed state and the average of the four initial hours of fasted state.

## Biochemical and hormonal plasma analyses

Plasma samples were obtained after a 6 h fasting period in EDTA-collection tubes. We centrifuged tubes at 1000 x g for 10 min and collected plasma, which was aliquoted and frozen in -80˚C until analysis. Glucose and total cholesterol concentrations were determined using a COBAS c111 Analyzer (Roche), following instructions from the manufacturer. Triglyceride concentrations were measured using a Triglyceride Colorimetric Assay Kit (DIASYS, Cat. No: 1 5760 99 10 021 R). Plasma insulin, adiponectin and leptin concentrations were determined using mouse/rat ELISA kits (ALPCO, CAT No: 80-INSRT-E01; 22-ADPRT-E01; 22-LEPMS-E01 respectively).

## Hepatic lipids analyses

Rat liver tissue (100 mg) was homogenized with 500 μL of 0.9% NaCl by a Kinematica Polytron Homogenizer. Then, using methanol and chloroform, lipids were extracted following Folch´s method [26]. The lipid extract was dried using a liquid nitrogen flow. Once we obtained the solid lipids, we redissolved them with 300 μL of Isopropanol/10% triton and determined triglycerides and total cholesterol using colorimetric assay Kits (DIASYS, Cat. No: 1 5760 99 10 021 R and 1 1350 99 10 021 R, respectively).

## Hepatic histological analyses

Liver tissues were fixed with 4% formaldehyde, dehydrated and embedded in paraffin. Subsequently, 5 μm sized tissue sections were cut and deparaffinized, and finally stained with hematoxylin and eosin (H&E) for morphological analysis. Additionally, snap-frozen samples were cryosectioned with a size of 10 μm, and subsequently stained with Oil Red O solution, for lipid content analysis. We analyzed five images per section and five sections from each rat liver.

## Hepatic gene expression analysis

Total RNA was isolated from 50 mg of frozen liver tissue using TRIzol reagent (Invitrogen) following manufacturer´s instructions. RNA quality and integrity was confirmed by spectrophotometry and agarose gel electrophoresis, respectively. Subsequently, total RNA (500 ng) was

used for reverse transcription to obtain cDNA using MultiScribe™ RT cDNA Synthesis Kit (Thermo Fisher Scientific). cDNA was obtained to determine gene expression by real time-PCR, using SYBR Green Master Mix (Roche) and the LightCycler 480 II System (Roche). To quantify relative expression, we used the Pfaffl method [27], utilizing two control genes with constitutive expression, including hypoxanthine phosphoribosyltransferase 1 (*Hprt*) and actin (*Actn*). We analyzed the expression of genes involved in cholesterol metabolism, including sterol regulatory element binding transcription factor 1 (*Srebf2*), 3-Hydroxy-3-Methylglutaryl-CoA reductase (*Hmgcr*), Low Density Lipoprotein receptor (*Ldlr*), *Lxrα* and cytochrome P450 family 7 subfamily A member 1 (*Cyp7a1*). The primers sequences for the analyzed genes are listed in S1 Table, all primers were used at a final concentration of 1 μM.

### Global DNA methylation and hydroxymethylation determination in liver

High quality DNA was isolated from 100 mg of liver tissue using phenol:chloroform:isoamyl technique to obtain 1 mg of DNA for subsequent Dot Blot Analysis [28]. For the Dot Blot, we denatured 500 μg of DNA with 0.1 M NaOH by heating 10 min at 95˚C. DNA samples were diluted and loaded in a nitrocellulose membrane of 0.45 μm. We used as controls unmethylated, methylated and hydroxymethylated cytosines from the Zymo Research DNA Standard Set (Cat. No: D5405), which we also denatured and loaded in the membrane. The membranes were incubated with specific antibodies against 5mC and 5hmC (Invitrogen, Cat no: MA5-24694 and MA5-695, respectively). The presence of methylated and hydroxymethylated cytosines was determined using a secondary anti-body signal and a chemiluminescent kit (Millipore, Immobilon), according to manufacturer's instructions. Chemiluminescence was quantified by ImageJ 1.53K (NIH) software.

### Statistical analyses

The results are presented as means ± SEM. Differences among groups were determined by one-way ANOVA and mixed-effects ANOVA, followed by Tukey´s post hoc tests. The effects of diet and age were considered statistically significant if $p < 0.05$. The ROUT method was used to identify outliers for all variables. Normal distribution was assessed by the Kolmogorov-Smirnov test, and the Kruskal-Wallis test was used for variables that did not follow a normal distribution, including plasma glucose and leptin levels, hepatic lipids, and the expression of the *Cyp7a1* and *Hmgcoar* genes. Statistical analyses were performed using Prism 6 (Graph-Pad).

## Results

### Prepubertal CD decreases fat mass accumulation and prevents lean mass loss in adulthood upon a chronic HFD

To evaluate whether a CD during prepuberty could decrease body weight in adulthood upon a chronic HFD, and whether a HFD in prepuberty could modify adult body weight upon a chronic CD, we assessed body weight longitudinally in rats fed according to our model of study (**Fig 1A**). We did not observe an effect of the prepubertal diet on weight at PND 220, since **C→HFD** and **HFD→C** had no statistical differences between them, or with the control groups, **CC** or **HH**. However, the slope of the weight trajectory and the final weight in adulthood of the **C→HFD** group was statistically similar to the **CC** group, while the **HH** group was not, implying that a prepubertal CD could have a mild effect protecting from chronic HFD induced excessive weight gain (**Fig 1B**). Regarding energy intake, groups fed with HFD had a

higher intake than groups fed with CD during prepuberty. Nevertheless, the energy intake during adulthood was similar among all groups (**Fig 1C**).

We next assessed whether the prepubertal CD or HFD could influence body composition in adulthood. **C→HFD** group had significantly ($p<0.05$) lower fat mass accumulation and a higher muscle mass percentage in adulthood than **HH** group. However, we observed that the slopes of the **C→HFD** group followed the pattern of the **HH** group, implying that the chronic diet leads to the same fat mass accumulation rate and lean mass loss, and thus, the prepubertal diet had no impact on the weight gain rate once exposed to a HFD. Additionally, the **HFD→C** group did not have a higher fat mass or lower lean mass than the **CC** group (**Fig 1D, 1E**). In summary, these results evidence that a prepubertal CD can attenuate body weight and fat mass accumulation, and lean mass loss despite a chronic HFD feeding in adulthood. However, the prepubertal diet had no impact on the rate of weight gain once exposed to either a HFD or a CD.

## Prepubertal CD improves metabolic flexibility in adulthood upon a chronic HFD

Since the **C→HFD** group had lower fat mass and higher lean mass than the **HH** group, we sought to observe whether prepubertal diet could influence energy metabolism by performing indirect calorimetry. As shown in **Fig 2**, the **C→HFD** group displayed a trend of higher $VO_2$ consumption than the **HH** group (**Fig 2A–2D**), which could be due to a higher energy expenditure [29]. Similar to body composition, the **HFD→C** group did not display any difference with the **CC** group. Now, regarding RER and, therefore, substrate utilization [29], we observed that the **C→HFD** group had a lower fasting RER compared to the **HH** group (**Fig 2E–2H**), indicating an increased and more efficient oxidation of fatty acids during fasting. Moreover, when we fed a CD, with the same amount of carbohydrates and lipids to all animals, we observed that the **C→HFD** group had a higher RER than the **HH** group, suggesting higher utilization and oxidation of glucose of the **C→HFD** group during the CD feeding test (**Fig 2H**), and therefore, an improved metabolic flexibility [29]. On the other hand, we observed that the **HFD→C** group had lower RER in the fed state while higher during fasting than the **CC** group, suggesting an alteration in the utilization of substrates during the fasted and fed states. Notably, the **C→HFD** group displayed a higher ΔRQ than the **HH** group in adulthood, while the **HFD→C** group displayed lower ΔRQ than the **CC** group (**Fig 2I**), demonstrating that a prepubertal CD or HFD influenced metabolic flexibility in adulthood [29, 30]. These findings suggest that a prepubertal CD could improve metabolic flexibility during adulthood.

## Prepubertal CD improves plasma leptin and cholesterol concentrations in adulthood upon a chronic HFD

We next determined biochemical and hormonal measurements in plasma. We observed that insulin have a tendency to be programmed by the prepubertal diet, since the **HFD→C** group displayed higher insulin concentrations than the **CC** group in adulthood, while **C→HFD** had lower insulin concentrations than the **HH** group (**Fig 3D**). Additionally, we did not observe an effect of the prepubertal diet on glucose (**Fig 3A**) or triglycerides (**Fig 3B**). Regarding adiponectin plasma concentrations, we observed that the levels of the adipokine increased during prepuberty, and that the CD led to a higher increase. However, we did not observed differences in adulthood among any groups (**Fig 3G and 3H**). Since the adipokines analyzed, including adiponectin and leptin, are influenced and correlate with fat mass [31], we normalized the concentrations of both adipokines with the percentage of fat mass of the total body weight. For adiponectin, we observed that concentrations were influenced by the diet consumed chronically. **CC** and **HFD→C** groups had similar adiponectin concentrations in adulthood, which

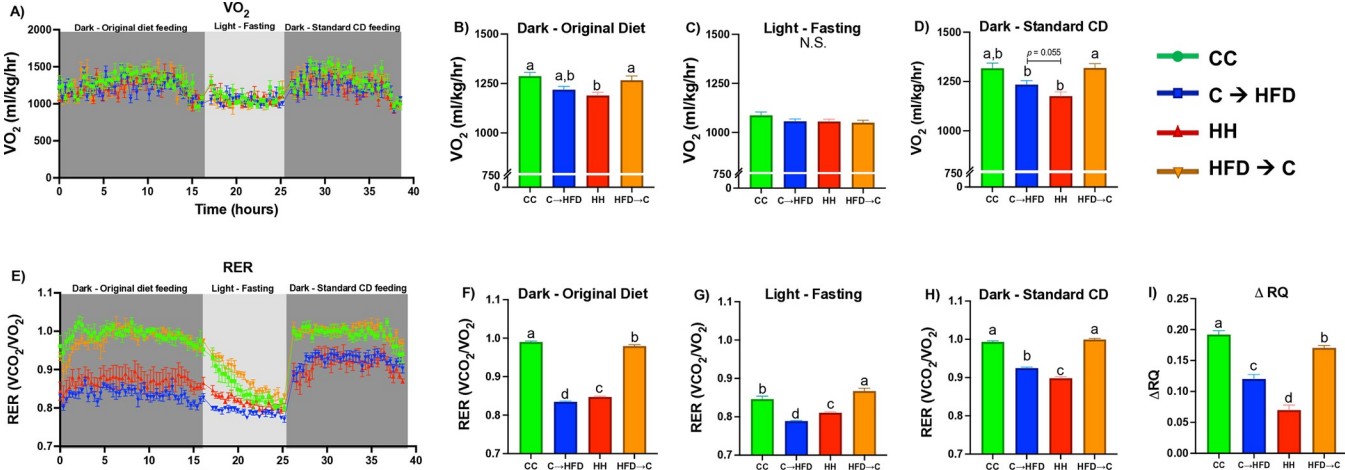

**Fig 2. A prepubertal CD maintains metabolic flexibility in adulthood despite a HFD in Sprague-Dawley male rats.** *(A)* Oxygen consumption (VO₂) from rats fed their original diet (OD) during dark, fasted during light and fed with control diet (CD) during dark final cycle. VO₂ average during OD feeding *(B)*, fasting *(C)*, and CD *(D)* for all groups. *(E)* Respiratory exchange ratio (RER), *(F-H)* RER average in same conditions as explained for VO₂, and *(I)* Delta respiratory quotient (ΔRQ) indicative of metabolic flexibility (Fed-state RER–fasting RER) all assessed by indirect calorimetry. Values are means ± SEMs (n = 5–6 rats per group in all panels). One-way ANOVA analyses were performed followed by Tukey´s post hoc tests. Letters indicate significant differences among groups, where: a > b > c > d (*p* < 0.05). Abbreviations: N.S. = no significance; RER = Respiratory exchange ratio; ΔRQ = Delta respiratory quotient; CC = Control Diet for 180 days; C→HFD = Control Diet for 10 days and HFD for 180 days; HH = HFD for 180 days; HFD→C = HFD for 10 days and CD for 180 days; CD = Control Diet.

were higher than those of the **HH** and **C→HFD** groups. Additionally, adiponectin levels decreased with age in all groups. Therefore, the prepubertal diet did not affect adiponectin concentrations in adulthood (**Fig 3G and 3H**). Notably, we observed that the **C→HFD** group had lower leptin and cholesterol concentrations than the **HH** group as observed in **Fig 3C and 3F**. Meanwhile, the **HFD→C** group had higher concentrations of cholesterol and leptin in adulthood compared to the **CC** group. Besides, the slope of the trajectory of cholesterol concentrations in plasma was significantly higher in **HFD→C** and **HH** groups compared to the **CC** group, while **C→HFD** and **CC** groups had significantly lower slope than **HH** group (**Fig 3C**). These results denote that the prepubertal diet had a programming effect on cholesterol and leptin concentrations during adulthood.

## Prepubertal CD protects against hepatic lipid accumulation in adulthood upon a chronic HFD

Since high cholesterol levels in adulthood are strongly associated with the development of cardiovascular events [32], and we observed that **C→HFD** group had significantly lower plasma cholesterol concentrations than the **HH group** (**Fig 3C**), we decided to analyze if the prepubertal CD could prevent hepatic lipid accumulation in adulthood. We observed that the **C→HFD** had lower hepatic triglycerides and cholesterol concentrations than the **HH** group (**Fig 4A and 4B**). However, the slopes of the trajectory of both lipids were similar between **C→HFD** and **HH** groups, and between **HFD→C** and **CC** groups. Furthermore, we analyzed the hepatic structure and lipid content by H&E and Oil Red O liver histology, respectively, to evaluate hepatic deterioration. The H&E histologic analysis displayed that the **C→ HFD** group had fewer vacuoles and higher cellular integrity than the **HH** group, while the **HFD→C** group did not display cellular alterations (**Fig 4C top panel**). Regarding the Oil Red O histology, we observed that the **C→HFD** group had lower hepatic lipid accumulation compared to the **HH** group (**Fig 4C lower panel**). Meanwhile, the **HFD→C** group had higher lipids when

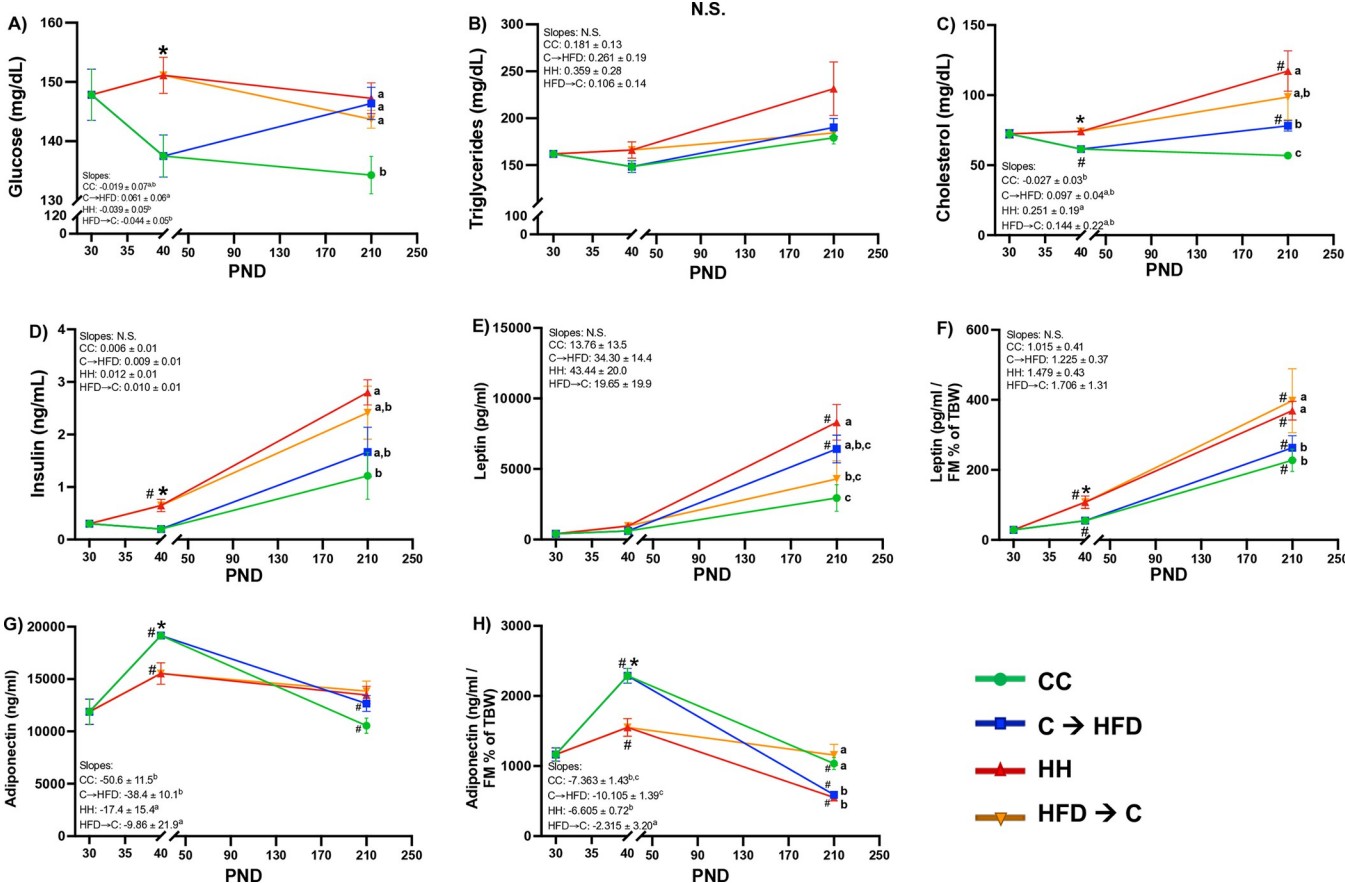

**Fig 3. A prepubertal CD decreases plasma leptin and cholesterol concentrations in adulthood despite a chronic HFD in Sprague-Dawley male rats.** *(A)* Glucose, *(B)* Triglycerides, *(C)* Cholesterol, *(D)* Insulin, *(E)* Leptin, *(F)* Leptin normalized, *(G)* Adiponectin and *(H)* Adiponectin normalized curves in plasma. Levels are assessed by colorimetric assays, Cobas or ELISA assays. Values are means ± SEMs (*n* = 5–6 rats per group in all panels). Mixed effects ANOVA analyses were performed followed by Tukey´s post hoc tests. Letters indicate significant differences among groups, where: a > b > c (*p* < 0.05), #: *p* < 0.05 compared to the respective previous time point, and *: *p* < 0.05 comparing CD vs. HFD in T1. Abbreviations: FM = Fat Mass; TBW = Total Body Weight; CC = Control Diet for 180 days; C→HFD = Control Diet for 10 days and HFD for 180 days; HH = HFD for 180 days; HFD→C = HFD for 10 days and CD for 180 days.

compared to the **CC** group (**Fig 4C lower panel**). Integrating the cholesterol concentrations in plasma and liver, and the hepatic lipid accumulation observed in the histologic images, the results suggest that the prepubertal CD or HFD diet programmed hepatic lipid metabolism, specifically cholesterol metabolism, despite the chronic HFD or CD, respectively.

## Prepubertal CD increased LXRα expression in adulthood upon a chronic HFD

Based on the observation that **C→HFD** group had lower plasma and hepatic cholesterol concentrations than **HH** group during adulthood, we wanted to understand if the prepubertal diet could affect the expression of enzymes involved in hepatic cholesterol metabolism, including synthesis, transport and excretion, during adulthood despite chronic HFD or CD. Hepatic cholesterol metabolism involves 3 main pathways that include: 1) Cholesterol *de novo* synthesis, where *Srebf2* is the transcription factor that activates the expression of the rate-limiting step enzyme for cholesterol synthesis, namely *Hmgcr* [33]. For both genes, we did not observe an influence or programming by the prepubertal diet. The **CC** and **HFD→C** groups had higher *Srebf2* and *Hmgcr* hepatic expression than the **HH** and **C→HFD** groups (**Fig 5A and 5B**). 2)

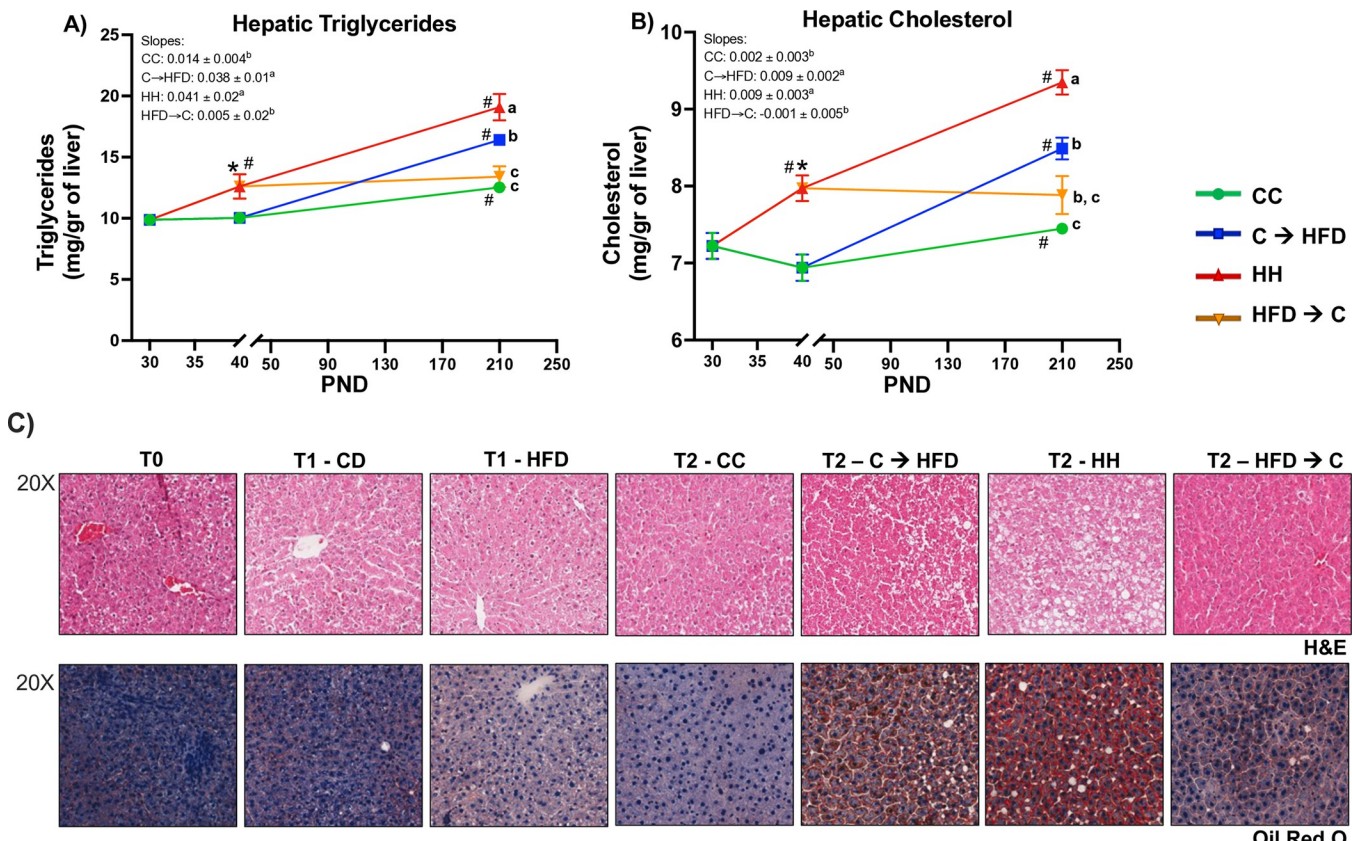

**Fig 4. A prepubertal CD decreases hepatic lipid accumulation in adulthood despite a chronic HFD in Sprague-Dawley male rats.** *(A)* Hepatic triglycerides and *(B)* hepatic cholesterol concentrations assessed by colorimetric assays. Values are means ± SEMs (*n* = 5–6 rats per group in all panels). Mixed effects ANOVA analyses were performed followed by Tukey´s post hoc tests. Letters indicate significant differences among groups, where: a > b > c (*p* < 0.05), #: *p* < 0.05 compared to the respective previous time point, and *: *p* < 0.05 comparing CD vs. HFD in T1. *(C)* Representative images of liver histology and hepatic lipid accumulation assessed by H&E (top panel) and Oil Red O staining (lower panel) respectively (20X). Abbreviations: H&E = Hematoxylin and Eosin; CC = Control Diet for 180 days; C→HFD = Control Diet for 10 days and HFD for 180 days; HH = HFD for 180 days; HFD→C = HFD for 10 days and CD for 180 days. T0 = PND30, T1 = PND40, T2 = PND210.

Cholesterol transport, were *Ldlr* codifies for the receptor that internalize LDL cholesterol inside the liver, and its expression is also regulated by *Srebf2* [33]. Notably, we observed that the **C→HFD** group had higher expression of *Ldlr* compared to the **HH** group (**Fig 5C**). And finally, 3) cholesterol excretion that occurs by its conversion to bile acids in the liver, were LXRα is the transcription factor that induces the expression of *Cyp7a1*, which is the rate-limiting step enzyme in charge of the conversion of cholesterol to bile acids for its later excretion [33]. The results showed that the **HFD→C** group had lower *Lxrα* expression than the **CC** group, while the **C→HFD** group was similar to all groups including the **CC** group (**Fig 5D**). Now, for *Cyp7a1*, there were no differences among groups (**Fig 5E**). These findings suggest that the increased expression of *Ldlr* and *Lxrα* in the **C→HFD** group, compared to **HH** group, may contribute to the lower plasmatic cholesterol levels by enhancing hepatic cholesterol internalization.

## Prepubertal CD programmed global DNA methylation and hydroxymethylation levels in liver in adulthood upon a chronic HFD

One of the mechanisms involved in metabolic programming during critical developmental periods, such as prepuberty, is DNA methylation and hydroxymethylation establishment [34].

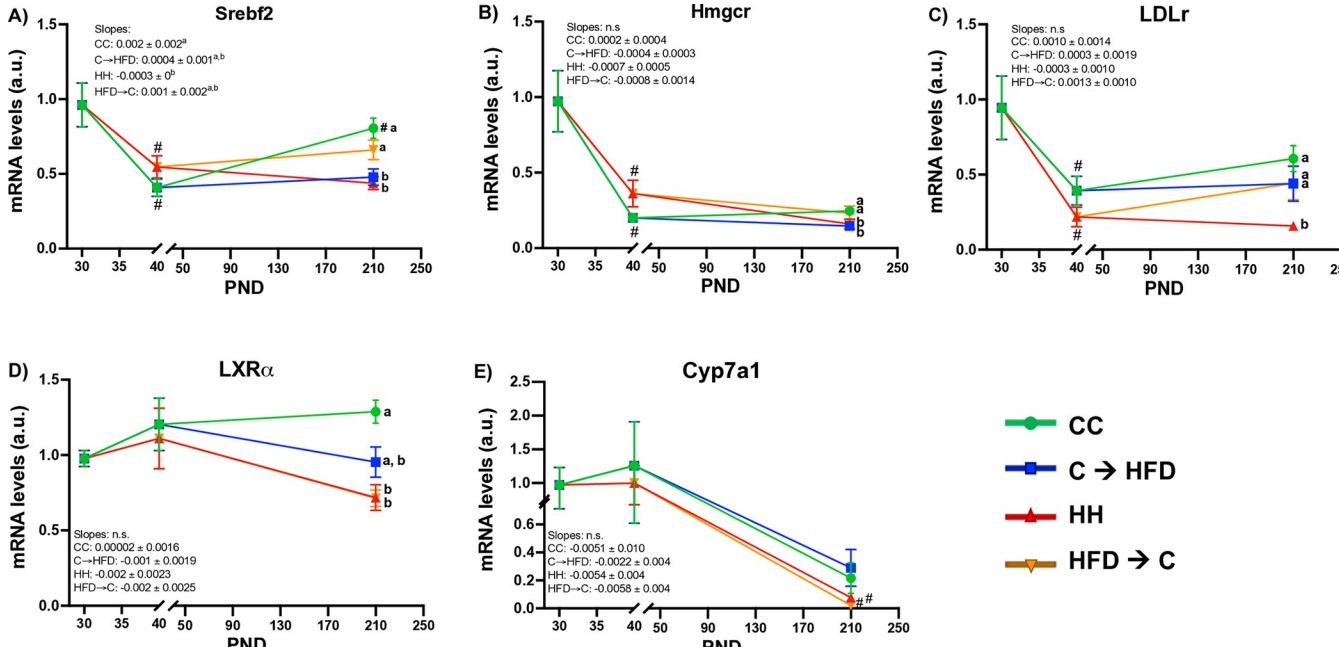

**Fig 5. A prepubertal CD increases *Ldlr* and *Lxrα* hepatic expression in adulthood despite a chronic HFD in Sprague-Dawley male rats.** mRNA levels assessed by real time-PCR of *(A)* of *Srebf2*, *(B) Hmgcr*, *(C)Ldlr*, *(D) Lxrα* and *(E) Cyp7a1* all of which are involved in hepatic cholesterol metabolism. Values are means ± SEMs (*n* = 5–6 rats per group in all panels). Mixed effects ANOVA analyses were performed followed by Tukey´s post hoc tests. Letters indicate significant differences among groups, where: a > b > c (*p* < 0.05) and #: *p* < 0.05 compared to the respective previous time point. *Srebf2* = Sterol regulatory element binding protein 2; *Hmgcr* = 3-hydroxy-3-methylglutaryl coenzyme A reductase; *Ldlr* = Low-density lipoprotein receptor; *Lxrα* = Liver-X-receptor alpha; *Cyp7a1* = cholesterol 7-alpha-monooxygenase or cytochrome P450 7A1; CC = Control Diet for 180 days; C→HFD = Control Diet for 10 days and HFD for 180 days; HH = HFD for 180 days; HFD→C = HFD for 10 days and CD for 180 days.

Moreover, studies have shown that the *Lxrα* promoter is susceptible to increased methylation during development, which can be maintained and influence the adult phenotype [23]. Therefore, we sought to determine whether the prepubertal diet could modify the global DNA methylation and hydroxymethylation profiles in the liver. Remarkably, the **C→HFD** group had lower methylation levels than the **HH** and **HFD→C** groups. Conversely, the **HFD→C** group increased their methylation levels compared to the **CC** group and to the same extent than the **HH** group (**Fig 6A**). Regarding 5hmC, we observed an inverse pattern, where the **HFD→C** and **HH** groups had lower levels compared to the **CC** group (**Fig 6B**). During prepuberty, only the **C→HFD** and **CC** groups increased their 5hmC levels, which were maintained by the **CC** group into adulthood, but decreased in the **C→HFD** group (**Fig 6B**). The **HFD→C** group never recovered their 5hmC levels, suggesting that 5hmC levels are established during prepuberty, but this establishment is disrupted by a HFD. These findings demonstrate that DNA methylation levels are maintained, and hydroxymethylation levels increase during prepuberty. Furthermore, a HFD consumed during this period increases DNA methylation and disrupts hydroxymethylation levels in adulthood, even when a CD is consumed afterward.

## Discussion

In this study, prepubertal CD feeding protected against the development of HFD-induced metabolic inflexibility, high cholesterol in plasma, hepatic lipid accumulation, decreased *Lxrα* expression, and decreased DNA hydroxymethylation levels despite a chronic HFD. Conversely, a HFD provided only during prepuberty was sufficient to decrease metabolic flexibility, increase serum cholesterol, and decrease hepatic *Lxrα* expression and global DNA

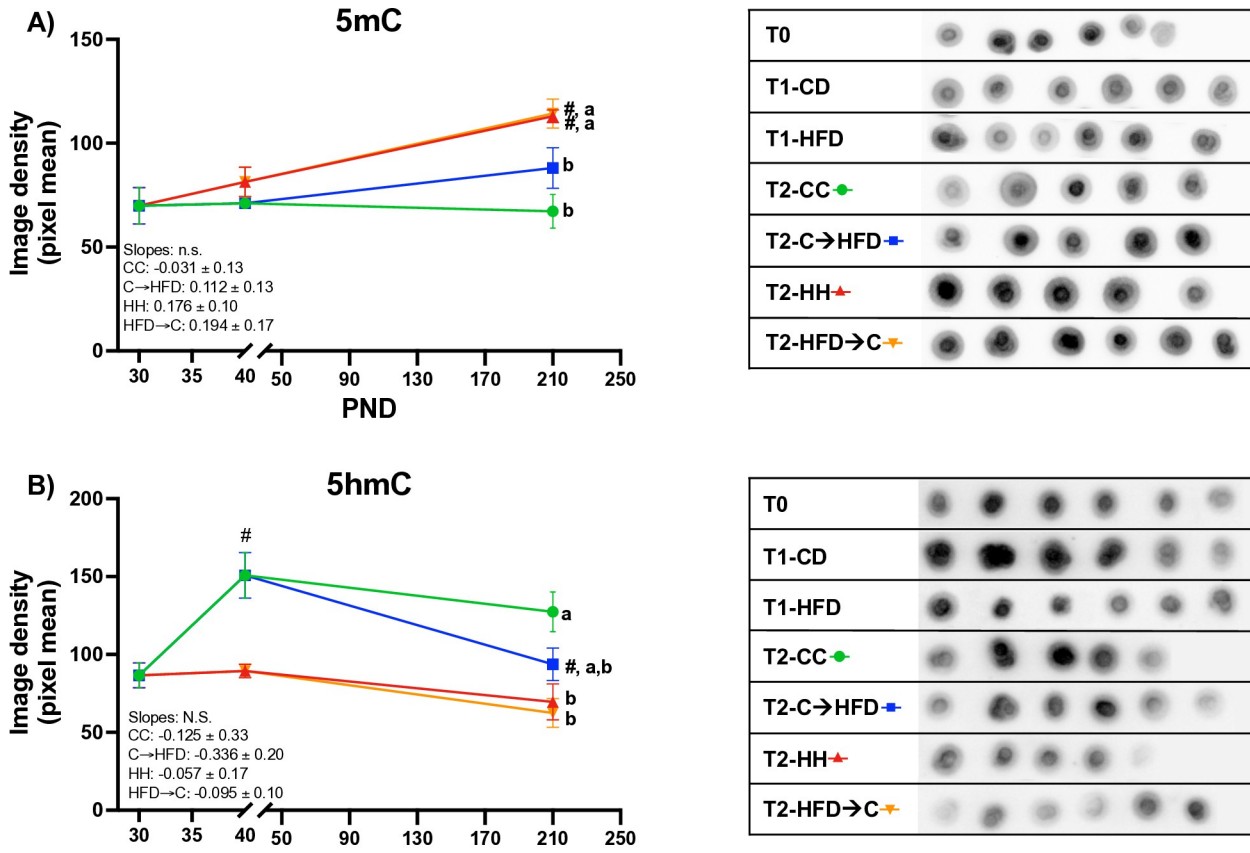

**Fig 6. A prepubertal CD decreases hepatic DNA methylation and increases hydroxymethylation in adulthood despite a chronic HFD in Sprague-Dawley male rats.** *(A)* Global methylation and *(B)* hydroxymethylation levels in hepatic DNA were assessed by Dot Blot and image density. Each dot represents an individual rat. Uncropped images are shown in S2 Fig. Values are means ± SEMs (n = 5–6 rats per group in all panels). Mixed effects ANOVA analyses were performed followed by Tukey´s post hoc tests. Letters indicate significant differences among groups, where: a > b > c (p < 0.05) and #: p < 0.05 compared to the respective previous time point. Abbreviations: 5mC = 5$^{th}$ methylated cytosine; 5hmC = 5$^{th}$ hydroxymethylated cytosine; CC = Control Diet for 180 days; C→HFD = Control Diet for 10 days and HFD for 180 days; HH = HFD for 180 days; HFD→C = HFD for 10 days and CD for 180 days; T0 = PND30, T1 = PND40, T2 = PND210.

hydroxymethylation, while increasing DNA methylation levels in the adult phenotype despite a chronic CD. Preventing cholesterol increases in the adult phenotype due to a prepubertal CD has significant implications, primarily due to the role of cholesterol in atherosclerosis development and, consequently, cardiovascular events [35].

Our study provides evidence showing that prepuberty is a critical window of development. Other studies showing similar results have described that prepubertal weight gain in SD male rats determined fat mass in adulthood [5], also that a chronic HFD initiated during prepuberty leads to increased leptin concentrations [6], impaired fertility, oxidative stress, and decreased testosterone [36, 37] in adulthood compared to controls in male rats. Additionally, Dinesh et al. [38] found that genetically obese male rats (WNIN) recovered their fertility rate only when subjected to a calorie-restricted diet starting in the prepubertal period rather than later. This was attributed to improved lower circulatory lipids and decreased oxidative stress [38]. Cheng et al. described that male SD rats developed obesity upon HFD feeding, but only when this started upon weaning at 21 days of age and not at 60 days of age. Therefore, younger rats were more susceptible to obesity induction, hypertension, hypercholesterolemia, and hepatic lipid deposition [39]. In contrast to the effects of a HFD, Rizzoto et al. demonstrated that

severe calorie restriction during prepuberty altered IGF concentrations and led to delayed puberty in male SD rats [21]. These findings support our observation that prepuberty is a critical developmental window influencing adult disease risk. However, these studies initiated dietary treatments at the onset of prepuberty and maintained them chronically. In contrast, we modified the chronic diet to assess its effect solely during this period. This strengthens the hypothesis that diet during this critical period has long-lasting effects. Oliveira-Ferreira et al., like us, provided a specific diet, protein-restricted compensated with higher carbohydrate content, exclusively during prepuberty, and also found that it could have a long-term impact on the adult phenotype despite a chronic CD. Specifically, the protein-restricted diet resulted in hypertension in the adult stage of male Wistar rats [40]. This demonstrates that not only an exclusive prepubertal HFD can have a long-term impact, but also a protein-restricted diet. However, Oliveira-Ferreira et al. did not describe the possible underlying mechanisms. Similarly, none of these studies have analyzed whether a control diet during prepuberty, a critical development period, could protect the metabolic phenotype in adulthood. Our study begins to address this deficiency, and explores how a short-term exposure to HFD during prepuberty might increase the susceptibility to metabolic dysregulation in adulthood.

Based on our results, this phenomenon could have been driven by changes in DNA methylation and hydroxymethylation, resulting in long-lasting alterations in gene expression. In fact, this hypothesis is further supported by a transgenerational study that showed male mice fed a protein-restricted diet increased expression of cholesterol biosynthesis genes in their offspring. This effect was linked to methylation changes, specifically at a putative enhancer of the peroxisomal proliferator-activated receptor alpha [41].

Finally, Holtrup et al. reported that a HFD exclusively during prepuberty caused increased fat mass, larger adipocytes, and higher leptin, insulin, and glucose concentrations in the adulthood of C57BL/6 mice despite a chronic CD [42]. The utilization of mice in this study notably indicates that the prepubertal dietary effect is conserved across both rats and mice, potentially extending to humans. The evaluation of exclusive exposures during prepuberty in humans and their long-term impact on health is limited and complex. However, a cohort study confirms that stimuli during prepuberty can impact the exposed offspring [43].

As we observed, a HFD either exclusively during prepuberty or throughout the entire study resulted in higher cholesterol concentrations. However, in contrast to human cholesterol metabolism, rats exhibit a higher rate of cholesterol excretion, attributed to increased bile acids excretion [44]. Therefore, rat models can be considered as more resistant to developing hypercholesterolemia due to a HFD. Nevertheless, many studies have demonstrated that a HFD can induce elevated cholesterol concentrations, potentially driven by hepatic lipid accumulation that alters cholesterol metabolism [45, 46]. Specifically, Magri-Tomaz et al. and St-Amand et al. demonstrated that two weeks of a HFD or Western Diet, respectively, caused changes in the expression of enzymes involved in cholesterol metabolism, including *Ldlr* expression [47, 48]. Additionally, a HFD can cause an increase in cholesterol hepatic accumulation and increased cholesterol concentrations in plasma by the rise of energy substrates that can increase the levels of cytoplasmic citrate and, therefore, acetyl-CoA, inducing cholesterol biosynthesis [49].

Our study found that increased cholesterol in plasma upon a HFD was associated with hepatic lipid accumulation and decreased *Lxra* and *Ldlr* expression, which suggests an underlying mechanism for increased plasmatic cholesterol.

In accordance with Reizel et al. [19], cholesterol concentrations in blood were one of the variables that were influenced by DNA demethylation. Upon analyzing gene expression and DNA methylation, we observed a relationship between prepubertal DNA methylation and hydroxymethylation status and the adult phenotype, suggesting epigenetic prepubertal

programming, which could lead to changes in the expression of *Lxrα* modulating hepatic cholesterol metabolism. Previous studies have shown that *Lxrα* is one of the genes affected by diet during critical windows of development, such as fetal development, and that its expression modifies cholesterol metabolism. The programming of *Lxrα* expression occurs through the establishment of DNA methylation [23, 50], a relevant process since alterations in *Lxrα* expression and cholesterol levels can impact cardiovascular health [51]. Since we also found that *Ldlr* expression was programed by prepubertal diet, and was associated to modified cholesterol concentrations, future studies should explore potential effects of the prepubertal diet on *Ldlr* methylation.

Now, given that we observed partial programming of hepatic DNA methylation and hydroxymethylation levels, it can be inferred that only certain regions maintained these epigenetic marks until adulthood. Therefore, considering that prepuberty is a crucial period for establishing DNA methylation patterns in the liver [52], additional studies should analyze into the specific impact of prepubertal diet on genome-wide DNA methylation status. This will help to understand all potential mechanisms that could contribute to the protective effect of a CD during prepuberty against HFD-induced obesity and metabolic complications.

Interestingly, the observed increase in 5hmC levels without a corresponding decrease in 5mC suggests that global DNA methylation and hydroxymethylation do not necessarily follow a straightforward inverse relationship. While 5hmC serves as an intermediate in DNA demethylation, it is often enriched in a tissue-specific manner, predominantly in regulatory regions of active genes. In contrast, 5mC is more broadly distributed across the genome, particularly in heavily methylated regions. According to Ivanov et al., 5hmC levels increase in the liver throughout development without a corresponding change in 5mC levels [53].

With regards to the underlying mechanism that could be behind the epigenetic programming of the resistance against developing metabolic alterations due to a prepubertal CD, or the increased susceptibility of metabolic alterations in response to a prepubertal HFD, we observed that during the prepubertal period, the HFD led to significantly higher glucose concentrations in plasma compared to a CD. This increase could have resulted in higher levels of intermediate metabolites necessary for one-carbon metabolism, thereby increasing the availability of methyl groups donated by S-adenosylmethionine (SAM). Considering that the prepubertal period is pivotal for DNA methylation changes [54], both the HFD and elevated glucose levels could have disrupted the establishment of DNA methylation and hydroxymethylation patterns. As noted by You et al. [55] and Tian et al. [56], a HFD can modify levels of methylation and hydroxymethylation by influencing the activity of the enzymes responsible for these processes, namely DNA-Methyltranferases (DNMTs) and Ten-Eleven translocases (TETs), respectively. Furthermore, the activity of these enzymes can compete to add either a methyl or a hydroxymethyl group to the DNA [57]. This aligns with our findings, where the prepubertal HFD prevented DNA hydroxymethylation in liver, and this was not recovered in adulthood. It was crucial to assess both methylation and hydroxymethylation since the former leads to gene repression, while the latter promotes expression.

Among the limitations of our study, although we adjusted the micronutrient composition of the HFD to ensure a similar intake among groups, subtle differences in the intake of 1-carbon donor metabolites, such as choline, may still be present. Given the role of 1-carbon metabolites, particularly S-adenosylmethionine (SAM), in DNA methylation, future studies should assess 1-carbon metabolites in plasma and liver to provide a clearer understanding of their potential impact on the adult phenotype and to clarify whether the observed differences in 5mC and 5hmC are due to micronutrient composition rather than fat content alone.

Also, we acknowledge that the experimental procedures primarily involve quantifying expression levels through real time-PCR, with protein levels not being assessed. However, it´s

important to note that the regulation of DNA methylation impacts RNA expression levels rather than protein levels. Regarding our Dot blot analysis aimed to determine global DNA methylation and hydroxymethylation, we recognize that there are more sensitive techniques available. Consequently, based on the findings of this study, sequencing techniques are planned for future studies.

Interestingly, our findings displayed that the prepubertal diet could influence adult metabolic flexibility. Considering the insulin and glucose concentrations in adulthood, it is possible that methylation could have altered the expression and activity of genes controlling the oxidation of substrates. For example, Jiang et al. [58] identified that a HFD increased the methylation and decreased hepatic hexokinase expression. Based on the metabolic flexibility findings of our study, as well as on leptin concentrations, a future aim for future studies is to determine how the prepubertal diet could modify the expression and methylation of key genes in adipose tissue, pancreas, and skeletal muscle.

Additionally, it is important to acknowledge that our observations in male rats might present a sexual dimorphism in female rats. For example, Reizel et al. [19] found that decreasing postnatal demethylation through a double knock-out of the TET2 and TET3 enzymes affected hepatic gene expression and glucose and cholesterol plasma levels in males but not in females, which was associated with testosterone release. Similarly, Tobi et al. [59] found that prenatal famine caused DNA methylation changes in a sex-dependent manner. Furthermore, prepuberty and the timing of puberty differ between males and females, and obesity affects this timing differently [60].

In conclusion, this study demonstrated that a prepubertal CD protects Sprague-Dawley male rats against elevated cholesterol levels, lower hepatic *LXRα* expression, and increased hepatic global methylation during adulthood despite a chronic HFD. Conversely, a prepubertal HFD decreased adult metabolic flexibility, increased serum cholesterol, and decreased *Lxrα* expression and global DNA hydroxymethylation, while also increasing DNA methylation levels despite a chronic CD. These findings suggest a potential role of prepubertal diet in preventing cardiometabolic complications during adulthood.

## Supporting information

**S1 Fig. Testosterone levels during prepuberty.** Values are means ± SEMs (n = 5–6 rats per group). ANOVA analyses were performed followed by Tukey´s post hoc tests. There were no statistical differences among groups. T0 = PND30; T1 = PND40; CD = Control Diet; HFD = High Fat Diet.
(PDF)

**S2 Fig. Raw images for Fig 6.** Uncropped images for dot blots in Fig 6.
(PDF)

**S1 Table. List of primer sequences for genes analyzed in this study.**
(DOCX)

## Acknowledgments

We thank Dr. Luis E. Gonzalez-Salazar for their technical support with COBAS measurements. Ana Aguilar-Lozano is a doctoral student from the Programa de Doctorado en Ciencias Biomédicas, Universidad Nacional Autónoma de México (UNAM) and has received CONAHCYT fellowship 692542.

## Author Contributions

**Conceptualization:** Ana Aguilar-Lozano, Berenice Palacios-González, Lilia G. Noriega.

**Formal analysis:** Ana Aguilar-Lozano, Berenice Palacios-González, Lilia G. Noriega.

**Funding acquisition:** Berenice Palacios-González, Lilia G. Noriega.

**Investigation:** Ana Aguilar-Lozano, Martha Guevara-Cruz, Alam Palma-Guzman, Lilia G. Noriega.

**Resources:** Berenice Palacios-González, Armando R. Tovar.

**Supervision:** Berenice Palacios-González, Lilia G. Noriega.

**Writing – original draft:** Ana Aguilar-Lozano.

**Writing – review & editing:** Berenice Palacios-González, Lilia G. Noriega.

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
