## [Decision Letter · Decision Letter 0]

23 Jul 2024

PONE-D-24-19131The type of diet consumed during prepuberty modulates plasma cholesterol, hepatic LXRα expression, and DNA methylation and hydroxymethylation during adulthood in male ratsPLOS ONE

Dear Dr. Noriega,

Thank you for submitting your manuscript to PLOS ONE. After careful consideration, we feel that it has merit but does not fully meet PLOS ONE’s publication criteria as it currently stands. Therefore, we invite you to submit a revised version of the manuscript that addresses the points raised during the review process..

Please follow the recommendations of the reviewers, and pay special attention to the changes in the presentation and description of results in the figures and text, and complete the references.

We look forward to receiving your revised manuscript.

Kind regards,

Víctor Sánchez-Margalet

Academic Editor

PLOS ONE

Journal Requirements:

6. Please amend either the title on the online submission form (via Edit Submission) or the title in the manuscript so that they are identical.

Reviewers' comments:

Reviewer's Responses to Questions

**Comments to the Author**

1. Is the manuscript technically sound, and do the data support the conclusions?

Reviewer #1: Yes

Reviewer #2: Yes

2. Has the statistical analysis been performed appropriately and rigorously? 

Reviewer #1: Yes

Reviewer #2: No

3. Have the authors made all data underlying the findings in their manuscript fully available?

Reviewer #1: Yes

Reviewer #2: Yes

4. Is the manuscript presented in an intelligible fashion and written in standard English?

Reviewer #1: Yes

Reviewer #2: Yes

5. Review Comments to the Author

Reviewer #1: The authors investigate prepuberty diets influence plasma cholesterol, hepatic LXR

expression, and DNA methylation and hydroxymethylation in male rats.

In the introduction the age of puberty for male rats should be described

Line 40: what tissues?

Line 41: how does Igf2 alter metabolic flexibility

Line 42: tends to be decreased - tends or is significantly decreased? ie is decreased

Line 46: reference

Line 49-what hepatic genes?

Line 50-what sex?

Line 73-what specific aspects of metabolic phenotype

All of the data (apart from histological and dot blot data) in figure 3-6 should be in bar graph with scatter plots

Line 333 reference

Line 335 define age of weaning

In the discussion, there should be some mention of any potential sex differences. Even though only males were studied in this project do the authors predict sex differences

Reviewer #2: This work brings a novelty to an important window of opportunity in development - prepuberty. Also, the study of HFD, that is widely used and studied, and it is well known to lead to chronic disease development throughout life. This paper is well-designed and described, addressing a great topic. However, some topics should be revised to better clarify the results and to be well understood, and then published.

1. In the introduction section. Second paragraph: when the gene is mentioned, it should be written in lowercase (Igf2), if it is a protein then uppercase (IGF2). Please review if it was correct.

2. In the introduction section. Second paragraph: is it really necessary to mention other types of diet instead of focusing only on the HFD which was the diet studied? References 20, 21, 22, 23, 24, and 25.

3. The study evaluated only male rats. Nowadays, we know that sex has an isolated effect on some metabolic results. When it is not possible to assess females either, it could be taken into consideration to discuss it in the appropriate section.

4. For statistical analyses, were there outliers removed? And the variables had normal distribution? This information could be placed in the section 2.8.

5. The meaning of “ * ” is not described in the legend of the figure.

6. It would be better if the group’s names were standardized (in text and figures are different from slopes).

7. The description of the results could be more concise, avoiding unnecessary mentions, because this could lead the reader to some misunderstanding. The results are confusing, maybe it would benefit from a rewrite.

For example, in 3.2: “Upon observing that a CD during prepuberty decreased fat mass and increased lean mass in adulthood despite a chronic HFD feeding”. It was not what is shown in the graph, the effect is despite chronic HFD when compared to HH. HFD→C seems to be similar to CC. Is this correct? So, if it is misunderstood maybe a rewrite could solve it.

8. All the abbreviations in the graphs must be in the legend.

9. Would be more appropriate to establish the comparisons in the results? Such as in the 3.5 section “Based on the lower plasma and hepatic cholesterol concentrations observed in the C→HFD group during adulthood” this means that is compared to all other groups?

10. If it is not statistically significant then I believe it is not right to state any conclusion. In the 3.5 section is said: “Now, for Cyp7a1, there was a trend that followed the same pattern of expression than Lxr�, where CC and C�HFD groups had higher Cyp7a1 expression than HH and HFD�C (Fig. 5D-E).”. Again, in lines 308-310. However, in the figure, there are no signals indicating it. Or maybe it is missing in the figures.

11. Across the results section are discussed hypotheses but without references. Would be more appropriate if those parts of the texts were moved to the discussion section? Then the results would be more concise and the discussion more robust. OR merge the two sections (results and discussion).

12. The sentence in lines 337-339 seems to be a little confusing, could the authors rewrite it for a better understanding?

13. In line 408, the abbreviation.

14. It is taken as a limitation the absence of other analyses in different tissues. Why is that missing in this paper? The results brought here can themself be explained and discussed, so perhaps it is not a limitation, but a future aim, some to be additional.

6. PLOS authors have the option to publish the peer review history of their article (what does this mean?). If published, this will include your full peer review and any attached files.

Reviewer #1: No

Reviewer #2: No

---

## [Author Response · Author response to Decision Letter 0]

6 Sep 2024

We appreciate the reviewers’ thoughtful comments and valuable contributions that have enabled to improve our manuscript titled “The Type of Diet Consumed During Prepuberty Modulates Plasma Cholesterol, Hepatic LXR� Expression, and DNA Methylation and Hydroxymethylation during Adulthood in Male Rats”. Below, please find our point-by-point response to the reviewers. We have highlighted in yellow the changes made in the revised manuscript. 

Reviewer 1:

 “The authors investigate prepuberty diets influence plasma cholesterol, hepatic LXR

expression, and DNA methylation and hydroxymethylation in male rats. 

We thank the reviewer for their time invested in revising our manuscript and for their valuable suggestions. A detailed point-by-point answer to their comments is provided below.

Comment 1: “In the introduction the age of puberty for male rats should be described.”

Response: We have included the age corresponding to prepuberty in male rats in line 52.

Comment 2: “Line 40: what tissues?”

Response: We have included in lines 39-40 that the mentioned study used peripheral blood cells. 

Comment 3: “Line 41: how does Igf2 alter metabolic flexibility”

Response: We have revised the manuscript to specify how IGF2 alters metabolic flexibility in line 42.

Comment 4: “Line 42: tends to be decreased - tends or is significantly decreased? ie is decreased”

Response: We have clarified that metabolic flexibility is decreased during obesity and high-fat overfeeding in line 43.

Comment 5: “Line 46: reference”

Response: We have revised the manuscript to include the corresponding reference now in line 47.

Comment 6: “Line 49-what hepatic genes?”

Response: We have included the hepatic genes that showed modified expression in the mentioned study in line 50-51.

Comment 7: “Line 50-what sex?”

Response: We have indicated that the results are from male rats in line 52.

Comment 8: “Line 73-what specific aspects of metabolic phenotype”

Response: We have included the specific aspects of the metabolic phenotype that were evaluated in line 75-76.

Comment 9: “All of the data (apart from histological and dot blot data) in figure 3-6 should be in bar graph with scatter plots”

Response: We have generated bar graphs with scatter plots for the suggested figures and made them available in Figshare, along with all the data included in this manuscript. However, we consider that our current figures are better suited to illustrate the effect over time, from the beginning of prepuberty to adulthood.

Comment 10: “Line 333 reference”

Response: We have included the corresponding reference in line 324.

Comment 11: “Line 335 define age of weaning”

Response: We have updated the manuscript to include the age of weaning in line 328.

Comment 12: “In the discussion, there should be some mention of any potential sex differences. Even though only males were studied in this project do the authors predict sex differences.”

Response: We have revised the manuscript to discuss potential sex differences.

Reviewer 2:

“This work brings a novelty to an important window of opportunity in development - prepuberty. Also, the study of HFD, that is widely used and studied, and it is well known to lead to chronic disease development throughout life. This paper is well-designed and described, addressing a great topic. However, some topics should be revised to better clarify the results and to be well understood, and then published.” 

We thank this reviewer for their time invested in revising our manuscript and for his/her positive comments and valuable suggestions. A detailed point-by-point answer to their comments is provided below.

Comment 1: “In the introduction section. Second paragraph: when the gene is mentioned, it should be written in lowercase (Igf2), if it is a protein then uppercase (IGF2). Please review if it was correct.”

Response: We have reviewed the manuscript and have used lower case italics for gene names and uppercase for protein names. In the particular case of IGF2 in line 42, we are indeed referring to the protein.

Comment 2: “In the introduction section. Second paragraph: is it really necessary to mention other types of diet instead of focusing only on the HFD which was the diet studied? References 20, 21, 22, 23, 24, and 25.”

Response: We agree with the reviewer that the manuscript introduction should focus solely on HFD. However, there is a limited number of studies that evaluate the postnatal effects of the type of diet during critical windows of development on adult DNA methylation patterns and/or phenotype. Therefore, we believe it is important to maintain these references in our introduction to highlight that the type of diet during critical windows of development can influence the adult phenotype.

Comment 3: “The study evaluated only male rats. Nowadays, we know that sex has an isolated effect on some metabolic results. When it is not possible to assess females either, it could be taken into consideration to discuss it in the appropriate section.”

Response: We completely agree with the reviewer. We have revised the manuscript to include a discussion on potential sex differences and how they may impact the metabolic results observed in our model.

Comment 4: “For statistical analyses, were there outliers removed? And the variables had normal distribution? This information could be placed in the section 2.8.”

Response: We apologize for the omission of this information. We have revised the statistical analyses section to clarify these points.

Comment 5: “The meaning of “ * ” is not described in the legend of the figure.”

Response: We have included “*”meaning in the figure legends.

Comment 6: “It would be better if the group’s names were standardized (in text and figures are different from slopes).”

Response: We apologize for the oversight. We have modified the figures to standardize groups´ names in text and figures. 

Comment 7: “The description of the results could be more concise, avoiding unnecessary mentions, because this could lead the reader to some misunderstanding. The results are confusing, maybe it would benefit from a rewrite. For example, in 3.2: “Upon observing that a CD during prepuberty decreased fat mass and increased lean mass in adulthood despite a chronic HFD feeding”. It was not what is shown in the graph, the effect is despite chronic HFD when compared to HH. HFD→C seems to be similar to CC. Is this correct? So, if it is misunderstood maybe a rewrite could solve it.”

Response: We agree with the reviewer and appreciate their observation. We have revised and updated the results to be more concise and clear, and we have included the appropriate references were necessary. 

Comment 8: “All the abbreviations in the graphs must be in the legend.”

Response: We have revised and updated the figure legends to include all abbreviations.

Comment 9: “Would be more appropriate to establish the comparisons in the results? Such as in the 3.5 section “Based on the lower plasma and hepatic cholesterol concentrations observed in the C→HFD group during adulthood” this means that is compared to all other groups?”

Response: We apologize for the oversight. We have updated the manuscript to address this issue on lines 267-268.

Comment 10: “If it is not statistically significant then I believe it is not right to state any conclusion. In the 3.5 section is said: “Now, for Cyp7a1, there was a trend that followed the same pattern of expression than Lxr�, where CC and C�HFD groups had higher Cyp7a1 expression than HH and HFD�C (Fig. 5D-E).”. Again, in lines 308-310. However, in the figure, there are no signals indicating it. Or maybe it is missing in the figures.”

Response: We apologize for the confusion. We have revised the text to remove any conclusions that were not supported by statistical significance.

Comment 11: “Across the results section are discussed hypotheses but without references. Would be more appropriate if those parts of the texts were moved to the discussion section? Then the results would be more concise and the discussion more robust. OR merge the two sections (results and discussion).”

Response: We apologize for the omission and appreciate the reviewer´s suggestion. We have included the appropriate references in the results section. Additionally, we have revised the manuscript to ensure a clear distinction between the presentation of results and their interpretation. 

Comment 12: “The sentence in lines 337-339 seems to be a little confusing, could the authors rewrite it for a better understanding?”

Response: We have rewritten the sentence now in line 330-331 for better understanding.

Comment 13: “In line 408, the abbreviation.”

Response: We have included the missing definition now in line 401.

Comment 14: “It is taken as a limitation the absence of other analyses in different tissues. Why is that missing in this paper? The results brought here can themself be explained and discussed, so perhaps it is not a limitation, but a future aim, some to be additional”

Response: We agree with the reviewer and have updated the discussion to frame the analysis of other tissues as a direction for future studies rather than as a limitation.

We hope that this new version will satisfy the reviewers and qualify our work for publication in PLOS One.

We look forward to hearing from you,

Sincerely,

Dr. Lilia Noriega, PhD.

---

## [Decision Letter · Decision Letter 1]

6 Oct 2024

PONE-D-24-19131R1The type of diet consumed during prepuberty modulates plasma cholesterol, hepatic LXRα expression, and DNA methylation and hydroxymethylation during adulthood in male ratsPLOS ONE

Dear Dr. Noriega,

Thank you for submitting your manuscript to PLOS ONE. After careful consideration, we feel that it has merit but does not fully meet PLOS ONE’s publication criteria as it currently stands. Therefore, we invite you to submit a revised version of the manuscript that addresses the points raised during the review process.If possible can you clarify the stage of puberty in the male, address the issue of vitamin status and clarify figure 6.

We look forward to receiving your revised manuscript.

Kind regards,

Christopher Torrens

Academic Editor

PLOS ONE

Journal Requirements:

Reviewers' comments:

Reviewer's Responses to Questions

**Comments to the Author**

1. If the authors have adequately addressed your comments raised in a previous round of review and you feel that this manuscript is now acceptable for publication, you may indicate that here to bypass the “Comments to the Author” section, enter your conflict of interest statement in the “Confidential to Editor” section, and submit your "Accept" recommendation.

Reviewer #1: (No Response)

Reviewer #2: All comments have been addressed

Reviewer #3: (No Response)

2. Is the manuscript technically sound, and do the data support the conclusions?

Reviewer #1: Yes

Reviewer #2: Yes

Reviewer #3: Yes

3. Has the statistical analysis been performed appropriately and rigorously? 

Reviewer #1: Yes

Reviewer #2: Yes

Reviewer #3: Yes

4. Have the authors made all data underlying the findings in their manuscript fully available?

Reviewer #1: Yes

Reviewer #2: (No Response)

Reviewer #3: Yes

5. Is the manuscript presented in an intelligible fashion and written in standard English?

Reviewer #1: Yes

Reviewer #2: Yes

Reviewer #3: Yes

6. Review Comments to the Author

Reviewer #1: The changes in the manuscript have improved it. No additional comments. The work is of interest to a wide audience

Reviewer #2: (No Response)

Reviewer #3: The study by Aguilar-Lozano et al explores the impact of diet around the time of puberty on adult metabolic health in rats. In this study, the authors specifically explore the association of feeding male rats either a control diet or a high fat diet between days 30 and 40 of age. The males are then either maintained on the same diet, or switched to the alternative diet, resulting in 4 groups. The specific aim was to see if a control diet around the time of puberty would prevent the development of metabolic disease in later life. In particular, the authors have focused on investigating hepatic function, histology, gene expression and DNA methylation.

On the whole, the manuscript is well-presented and well-written. However, I do have three main comments that need addressing.

The first is that the diets appear to have different levels of vitamins, minerals and 1-Carbon metabolites and carriers such as choline. As the authors are feeding these diets to their rats for up to 180 days, would they anticipate that the differences in 5mC and 5hmC are due, in part, to the differing levels of 1-Carbon metabolites within their diets? Also, do the rats eat the same weight of the high fat diet per day as the control diet. Again, as the diets are not balanced for the micro-nutrient levels, if the rats consume a different weight of the diets then would they be consuming differing levels of 1-Carbon metabolites and so be affecting their 5mC and 5hmC levels. As such, are the differences that are reported not due to the high levels of fat, per say, but just due to the different micro-nutrient compositions. This possibility needs to be addressed within the discussion.

My second question is, how much variability in the start of puberty is there between the males? As the authors have started all the males on their diets from 30 days of age, could some males be closer to puberty than others and so could they be affected differently when placed on their respective diet? Could the authors have defined their pubertal status in a more precise way through hormonal assessment?

My final comment relates to Figure 6. I find this figure a little confusing as it is not clear which group is represented at T0 in each of the dot blots. Also, does T0 equate to the start of the dietary challenges (i.e. 30 days of age), and so T1 is 40 days and T2 210 days? If so, why not use the days as T0, T1 and T2 are not used elsewhere in the manuscript. On the dot blot for 5hmC, should 'T1-C' be 'T1-CC'? Finally, i find it interesting that there is a significant increase in 5hmC in the CC and C-HFD groups at day 40, but no difference in levels of 5mC at the same time in the same animals. Would the authors anticipate that a significant increase in 5hmC would suggest a shift towards a reduction in 5mC levels as 5hmC is an intermediate state in a shift towards demethylation? I would welcome the authors comments on the lack of connection.

7. PLOS authors have the option to publish the peer review history of their article (what does this mean?). If published, this will include your full peer review and any attached files.

Reviewer #1: No

Reviewer #2: No

Reviewer #3: No

---

## [Author Response · Author response to Decision Letter 1]

3 Nov 2024

We thank the reviewer for their time invested in revising our manuscript and for their valuable suggestions. A detailed point-by-point answer to their comments is provided below.

Comment 1: “The first is that the diets appear to have different levels of vitamins, minerals and 1-Carbon metabolites and carriers such as choline. As the authors are feeding these diets to their rats for up to 180 days, would they anticipate that the differences in 5mC and 5hmC are due, in part, to the differing levels of 1-Carbon metabolites within their diets? Also, do the rats eat the same weight of the high fat diet per day as the control diet. Again, as the diets are not balanced for the micro-nutrient levels, if the rats consume a different weight of the diets, then would they be consuming differing levels of 1-Carbon metabolites and so be affecting their 5mC and 5hmC levels. As such, are the differences that are reported not due to the high levels of fat, per se, but just due to the different micro-nutrient compositions. This possibility needs to be addressed within the discussion.”

Response: 

Food intake in grams decreases in high-fat diet (HFD) groups due to the diet´s higher energy density. As a result, we adjusted the vitamin, mineral, L-cystine, and choline content in HFD diets, as is commonly done in studies using AIN93M as a control diet, to account for reduced food intake. While we cannot fully rule out the influence of dietary 1-carbon metabolites on 5mC and 5hmC levels without specific assessments, we have controlled for this to the extent possible by balancing these micronutrients in relation to intake. 

However, we recognize that a direct assessment of 1-carbon metabolites in plasma and liver could provide further insight into any differences in 5mC and 5hmC levels. We have addressed this issue into the discussion on line 420 to 426. 

Comment 2: “My second question is, how much variability in the start of puberty is there between the males? As the authors have started all the males on their diets from 30 days of age, could some males be closer to puberty than others and so could they be affected differently when placed on their respective diet? Could the authors have defined their pubertal status in a more precise way through hormonal assessment?

Response: 

To account for variability in pubertal timing, we measured testosterone levels at postnatal day 30 (T0) and postnatal day 40 (T1) in both diet groups to confirm that all animals were still in the prepubertal stage. The prepubertal stage typically corresponds to testosterone concentrations bellow 2.1 ng/mL (Lee et al). In our study, testosterone levels averaged 0.336 ng/mL (95% CI: 0.233 to 0.439), and no significant variability was observed among subjects within the same group. This assessment confirmed that none of the animals had reached puberty by these time points. Although natural biological variability exists, our hormonal assessment provided a reliable indicator of prepubertal status. We have added this information to the methods section at line 92.

Lee VW, de Kretser DM, Hudson B, Wang C. Variations in serum FSH, LH and testosterone levels in male rats from birth to sexual maturity. J Reprod Fertil. 1975 Jan;42(1):121-6. doi: 10.1530/jrf.0.0420121. PMID: 1110463.

Comment 3: “My final comment relates to Figure 6. I find this figure a little confusing as it is not clear which group is represented at T0 in each of the dot blots. Also, does T0 equate to the start of the dietary challenges (i.e. 30 days of age), and so T1 is 40 days and T2 210 days? If so, why not use the days as T0, T1 and T2 are not used elsewhere in the manuscript. On the dot blot for 5hmC, should 'T1-C' be 'T1-CC'? Finally, I find it interesting that there is a significant increase in 5hmC in the CC and C-HFD groups at day 40, but no difference in levels of 5mC at the same time in the same animals. Would the authors anticipate that a significant increase in 5hmC would suggest a shift towards a reduction in 5mC levels as 5hmC is an intermediate state in a shift towards demethylation? I would welcome the authors comments on the lack of connection.

Response:

We apologize for any confusion. We have clarified the meaning of T0, T1, and T2 in the figure legends for Figures 1, 4 and 6. T0, T1, and T2 represent the time points at which measurements were performed, specifically:

• T0 = Postnatal day (PND) 30,

• T1 = PND 40,

• T2 = PND 210.

In other figures, we opted to display postnatal days on the y-axis for clarity. In the dot blot, however, we used the combination of the time point (T0, T1, T2) and the diet group to provide specific context for each group´s timing and diet.

Regarding the labeling issue in the dot blot, we have corrected “T1-C” to “T1-CD” to align with the description in Figure 1A and the methods section, where “CC” refers to control diet throughout the study. Thus, "T1-CD" specifically represents the control diet group at day 40.

Concerning the increase in 5hmC levels without an associated decrease in 5mC, it is important to note that we assessed global rather than site-specific DNA methylation and hydroxymethylation. Thus, while 5hmC serves as an intermediate in demethylation, this does not necessarily lead to a direct inverse relationship with 5mC on a global level. The majority of CpG sites in the genome remain methylated, whereas hydroxymethylation is typically enriched in regulatory regions of actively transcribed genes. Based on previous findings (Reizel et al., reference 19), we expected that hydroxymethylation could increase as part of the demethylation process, though global 5mC levels might remain stable. We have incorporated this explanation into the discussion on line 396 and included references from Charlton et al. (reference 57) and Ivanov et al. (reference 53) to support this interpretation.

---

## [Editor Report · Decision Letter 2]

22 Nov 2024

The type of diet consumed during prepuberty modulates plasma cholesterol, hepatic LXRα expression, and DNA methylation and hydroxymethylation during adulthood in male rats

PONE-D-24-19131R2

Dear Dr. Noriega,

We’re pleased to inform you that your manuscript has been judged scientifically suitable for publication and will be formally accepted for publication once it meets all outstanding technical requirements.

Kind regards,

Christopher Torrens

Academic Editor

PLOS ONE

---

## [Editor Report · Acceptance letter]

14 Jan 2025

PONE-D-24-19131R2 

PLOS ONE

Dear Dr. Noriega, 

I'm pleased to inform you that your manuscript has been deemed suitable for publication in PLOS ONE. Congratulations! Your manuscript is now being handed over to our production team.

Kind regards, 

on behalf of

Dr. Christopher Torrens 

Academic Editor

PLOS ONE